# Adverse drug events leading to emergency department visits: A multicenter observational study in Korea

**Min-Gyu Kang**[1], **Ju-Yeun Lee**[2], **Sung-Il Woo**[3], **Kyung-Sook Kim**[4], **Jae-Woo Jung**[5]*, **Tae Ho Lim**[6], **Ho Joo Yoon**[7], **Chan Woong Kim**[8], **Hye-Ran Yoon**[9], **Hye-Kyung Park**[10], **Sang-Heon Kim**[7]*

**1** Department of Internal Medicine, Subdivision of Allergy, Chungbuk National University Hospital, Cheongju, Korea, **2** College of Pharmacy and Research Institute of Pharmaceutical Sciences, Seoul National University, Seoul, Korea, **3** Department of Pediatrics, Subdivision of Allergy, Chungbuk National University Hospital, Cheongju, Korea, **4** Department of Pharmacy, Chungbuk National University Hospital, Cheongju, Korea, **5** Department of Internal Medicine, Chung-Ang University College of Medicine, Seoul, Korea, **6** Department of Emergency Medicine, Hanyang University College of Medicine, Seoul, Korea, **7** Department of Internal Medicine, Hanyang University College of Medicine, Seoul, Korea, **8** Department of Emergency Medicine, Chung-Ang University College of Medicine, Seoul, Korea, **9** College of Pharmacy, Duksung Women's University, Seoul, Korea, **10** Department of Internal Medicine, Pusan National University School of Medicine, Busan, Korea

* sangheonkim@hanyang.ac.kr (SHK); jwjung@cau.ac.kr (JWJ)

**Data Availability Statement:** The data in this study cannot be shared publicly because of the limigations imposed by the study's ethics approval. Access to the de-identified data are available for

## Abstract

Adverse drug events are significant causes of emergency department visits. Systematic evaluation of adverse drug events leading to emergency department visits by age is lacking. This multicenter retrospective observational study evaluated the prevalence and features of adverse drug event-related emergency department visits across ages. We reviewed emergency department medical records obtained from three university hospitals between July 2014 and December 2014. The proportion of adverse drug events among total emergency department visits was calculated. The cause, severity, preventability, and causative drug(s) of each adverse drug event were analyzed and compared between age groups (children/adolescents [<18 years], adults [18–64 years], and the elderly [≥65 years]). Of 59,428 emergency department visits, 2,104 (3.5%) were adverse drug event-related. Adverse drug event-related emergency department visits were more likely to be female and older. Multivariate logistic regression analysis revealed that compared to non- adverse drug event-related cases, adverse drug event-related emergency department visitors were more likely to be female (60.6% vs. 53.6%, p<0.001, OR 1.285, 95% CI 1.025–1.603) and older (50.8 ± 24.6 years vs. 37.7 ± 24.4 years, p<0.001, OR 1.892, 95% CI: 1.397–2.297). Comorbidities such as diabetes, chronic kidney disease, chronic liver disease, and malignancies were also significantly associated with adverse drug event-related emergency department visits. Side effects were the most common type of adverse drug events across age groups, although main types differed substantially depending on age. Serious adverse drug events, hospitalizations, and adverse drug event-related deaths occurred more frequently in the elderly than in adults or children/adolescents. The proportion of adverse drug event-related emergency department visits that were preventable was 15.3%. Causative drugs of adverse drug

researchers who meet the criteria for access to confidential data. Request for data access should be directed to the institutional review boards of Hanyang University Hospital (irb@hanyang.ac.kr), Jungang University Hospital (irb@caumc.or.kr) and Chungbuk National University Hospital (irb@cbnuh.or.kr).

**Funding:** This research was supported by a grant for the Korea Health Technology R&D Project through the Korea Health Industry Development Institute (KHIDI), funded by the Ministry of Health & Welfare, Republic of Korea (HI19C0218); the National Research Foundation of Korea (NRF) grant funded by the government of Korea (2020R1F1A1069087); and a research grant from the Korea Institute of Drug Safety & Risk Management (2015-0002). The funders had no role in study design, data collection and analysis, decision to publish, or preparation of the manuscript. They encouraged publication after the decision to publish was made by the authors.

**Competing interests:** The authors have declared that no competing interests exist.

events varied considerably depending on age group. Adverse drug event features differ substantially according to age group. The findings suggest that an age-specific approach should be adopted in the preventive strategies to reduce adverse drug events.

## Introduction

The increase in the prevalence of chronic diseases and the development of therapeutic drugs has resulted in increased drug exposure in the population [1]. The increase in drug exposure has led to a rapid increase in the occurrence of adverse drug events (ADEs) [2]. In the United States (US), approximately 2.2 million people have been hospitalized due to ADEs [3]. Indeed, ADE is the fourth most common cause of death, resulting in 100,000 deaths annually [3]. ADE is a significant medical issue directly related to patients' life and socioeconomic factors.

Although most drugs are consumed outside hospitals, ADEs occurring in the outpatient setting are less extensively monitored than those occurring in hospitals. Monitoring of ADEs, which lead to emergency department (ED) visits, is a suitable approach to evaluate the occurrence of ADEs in outpatient settings [4]. Patients visit the ED when they experience serious ADEs in outpatient settings. In addition, since the causal relationships between drug administration and ADEs are relatively clear, it is easier to diagnose ADEs accurately [5]. Several studies have evaluated the prevalence and epidemiologic features of ADE-related ED visits by reviewing the medical records of patients visiting the ED [6–9]. Most studies were conducted in a single-center or a limited number of hospitals, with small sample size. Therefore, estimations of the prevalence and characteristics of ADE-related ED visits remain limited. Moreover, several reports based on nationwide databases or claims systems have been conducted. Cohen et al. analyzed ADE-related diagnostic codes in the National Electronic Injury Surveillance System-Cooperative Adverse Drug Event Surveillance (NEISS-CADES) system and evaluated the national estimates of ADE-related ED visits in children [10]. These studies have the advantage of estimating the prevalence at a nationwide level. However, considering that the negative predictive value of the ADE-related diagnostic code is low, these approaches may underestimate the prevalence of ADE-related ED visits in real life. In addition, it is difficult to evaluate the epidemiologic factors of ADE-related ED visits using these approaches.

Although ADE-related ED visits are a critical issue related to medication safety, the incidence, and epidemiologic characteristics of ADEs leading to ED visits have not been sufficiently studied. To estimate the true prevalence and the epidemiologic features of ADE-related ED visits, we conducted a multicenter pharmacoepidemiologic study based on a standardized protocol with many subjects who visited EDs at university hospitals.

## Materials and methods

### Study design and protocol

In this multicenter retrospective observational study (PERADE), three tertiary university hospitals, namely, Hanyang University Hospital, Chung-Ang University Hospital in Seoul, and Chungbuk National University Hospital in Cheongju, Korea, participated. These hospitals share many common features. First, all three hospitals were designated as Regional Emergency Medical Centers by the Ministry of Health and Welfare of Korea. The National Emergency Department Information System (NEDIS), which is a representative nationwide system that collects clinical and administrative data on patients who visit the ED in Korea, has been

implemented in all three hospitals. The NEDIS database contains key clinical information such as demographics, clinical information, initial vital signs, initial diagnosis, treatment outcomes, and dispositions, enabling each hospital's researchers to conduct the evaluation based on the same data format. Further, each hospital actively operates the Regional Pharmacovigilance Centers under the supervision of the Korea Institute of Drug Safety and Risk Management (KIDS) and the Korea Ministry of Food and Drug Safety. These hospitals operate similar pharmacovigilance systems and ADE-reporting systems to the KIDS.

To minimize heterogeneity in assessing ADEs between organizations and researchers, we generated a standardized written study protocol. We specified the purpose and outcomes of the study, the definition of ADEs, inclusion or exclusion criteria for cases, and outcomes for analyses. The ongoing status and preliminary results of the study were shared with other hospitals through regular research meetings to ensure that each institution conducted evaluations based on the same protocols, and consequently reduced the heterogeneity of reviewing processes between hospitals. Using a secured web-based electronic case report form (e-CRF), we stored and managed the data from the patients who visited the ED. Data quality management was performed by reducing errors, duplication, and omission of essential items and increasing fidelity.

## Subjects and case definition of ADEs

The study subjects were patients who visited the EDs of three hospitals over a 6-month study period, from July 2014 to December 2014. We thoroughly evaluated the study participant's electronic medical records (EMRs) during ED visits as well as for 1 year before and after ED visits. Fully anonymized information on the study subjects was extracted from electronic medical records (EMRs) and the NEDIS database of each hospital and uploaded on the e-CRF. An ADE was defined as untoward and unintended events arising from the use or misuse of medications. Using a two-step approach, detailed information regarding each ED visit was reviewed to determine ADEs based on information from the e-CRF and EMRs. In the first review, each case was screened and categorized as low, intermediate, or high possibility of ADE. Subsequently, pharmacovigilance and allergy specialists confirmed ADEs from intermediate or high possibility of ADEs by assessing causality categories based on the World Health Organization-Uppsala Monitoring Center (WHO-UMC) criteria [11]. ADE cases scored as certain, probable, or possible were included as ADEs, whereas the cases with unlikely, unclassified, or unclassifiable causalities were excluded. For each ADE, the diagnosis was coded according to the Korean Classification of Disease version-6 (KCD-6), and the name of the ADE was coded according to the preferred terms (PTs) or included terms (ITs) based on the World Health Organization-Adverse Reaction Terminology (WHO-ART) version 092 [12]. Culprit medications were coded to the WHO Anatomical Therapeutic Classification (ATC) system.

This study was approved by the Institutional Review Boards of Hanyang University Hospital (IRB 2015-08-004), Chung-Ang University Hospital (IRB C2015174-1632), and Chungbuk National University Hospital (IRB 2015-06-018). The ethics committee from each hospital waived the need for informed.

## Main outcomes of ADEs

As major outcomes of each ADE case, we determined the causes (mechanisms), severity, preventability, and causative drugs based on the standardized protocols of this study. Types of ADE were classified as follows: side effect, overdose, secondary effect, drug interaction, allergy, or non-allergic hypersensitivity reaction. Additionally, ADEs related to poor compliance, dose reduction discordant with physicians' opinion, or discontinuation of therapy were included in

this analysis [13]. ED visits associated with intentional overdose, drug abuse, and/or suicidal attempts were excluded from this study. The severity of ADEs was categorized into mild, moderate (needed treatment but recovery was possible), or severe (needed hospitalization or possibility of disability) [14]. The suspected ADEs were further analyzed for assessing the preventability by Schumock and Thornton algorithm [15]. Preventability of ADEs was defined when meeting any one of the following: drug inappropriate for a clinical condition; inappropriate dose, route, or frequency of administration; required therapeutic drug monitoring or lab test not performed; and history of allergies or hypersensitivity, drug interaction, toxic drug levels, and poor compliance.

## Statistical analysis

All statistical analyses were performed using IBM Statistical Product and Service Solutions (SPSS Statistics for Windows, version 22.0; IBM Corp., Armonk, NY, USA). A two-sided p-value <0.05 was considered statistically significant. The characteristics of the study population were summarized as both numbers and percentages for categorical variables and as mean ± SD for continuous variables. Chi-square statistics and independent t-tests were used to evaluate the differences between baseline characteristics of groups or outcomes in this study. We estimated 95% confidence intervals (CIs) using logistic regression for binary outcomes.

# Results

## Characteristics of ADE-related ED visits

A total of 59,428 cases of ED visits were recorded in three hospitals during the study period (from July 2014 to December 2014). Of these, 2,104 cases (3.5%) were identified as ADE-related ED visits. An estimate of 35.4 (95% CI, 33.8–36.6) per 1,000 ED visits during the study period was recorded. The mean age of patients with ADE-related ER visits was 50.8±24.6 years (Table 1). Compared with non-ADE-related cases, ADE-related ED visits were more likely to be female (60.6% vs. 53.6%, p<0.001) and older (50.8±24.6 years vs. 37.7±24.4 years, p<0.001). There were more elderly subjects for ADE-related ED visits than for non-ADE causes (30.1% vs. 16.8%, p<0.001). The numbers and proportion of ADE-related ED visit increased with age, with the highest value in subjects aged 70–74 years old (8.5%) (Fig 1). After ED visits, hospitalizations were more frequently required for ADE-related ED visits than for

**Table 1. Characteristics of the study subjects.**

| | ADE | Non-ADE | *P*-value |
|---|---|---|---|
| | **(n = 2,104)** | **(n = 57,324)** | |
| **Female** | 1,275 (60.6%) | 30,252 (53.6%) | <0.001 |
| **Age, years** | 50.8 ± 24.6 | 37.7 ± 24.4 | <0.001 |
| **Age group** | | | <0.001 |
| Children/adolescents (<18 years) | 242 (11.5%) | 13,070 (22.8%) | |
| Adults (18–64 years) | 1,088 (58.4%) | 34,599 (60.4%) | |
| Elderlies (≥65 years) | 774 (30.1%) | 9,655 (16.8%) | |
| **Hospitalization** | 613 (29.1%) | 12068 (21.1%) | <0.001 |
| **Duration of hospitalization, days** | 13.6 ± 26.0 | 12.0 ± 20.6 | 0.502 |
| **Death** | 24 (1.1%) | 768 (1.3%) | 0.490 |

ADE, adverse drug event; ED, emergency department

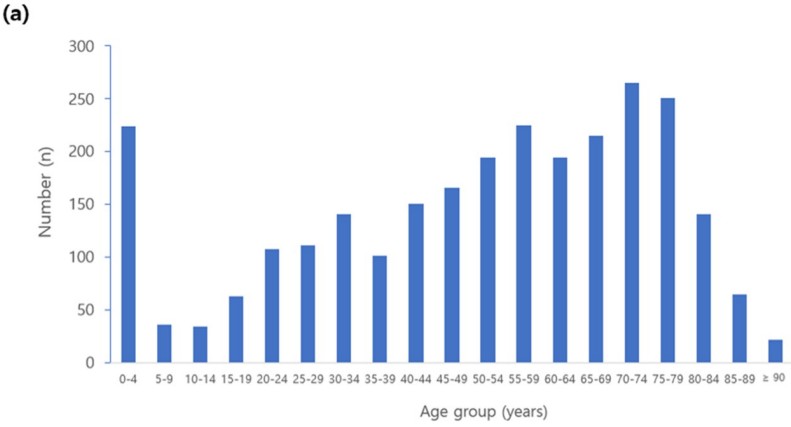

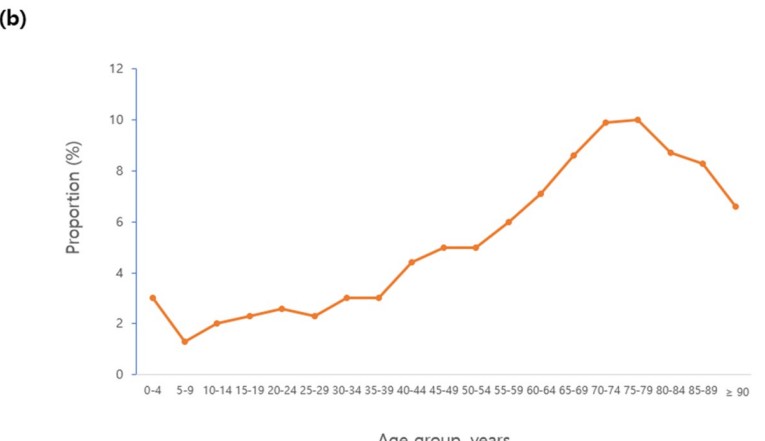

**Fig 1. ADE-related ED visits according to age.** (a) Case numbers of ADE-related ED visit according to age. (b) Proportions of ADE-related ED visits among total ED visits according to age.

non-ADE-related visits (29.1% vs. 21.1%, p<0.001). However, hospital days and mortality after ED visits were not significantly different between ADE- and non-ADE-related visits.

## Types and severity of ADEs

Table 2 presents the types (mechanisms) and severity of ADE-related ED visits, and comparisons according to age group: children and adolescents (<18 years), adults (18–64 years), and the elderly (≥65 years). Regarding types of ADEs, side effects (72.5%) were the most common cause, followed by allergy (13.6%), overdose (7.3%), non-allergic hypersensitivity reaction (3.3%), secondary effect (2.0%), and drug interaction (0.9%).

Analysis of the types of ADEs according to age group revealed that overdose was more frequently observed in elderly patients (11.2%) than in adults (5.8%) and children/adolescents (2.2%, p<0.001). However, allergies were more common in children/adolescents (27.1%) than in adults (15.3%) and the elderly (5.2%, p<0.001).

The severity of ADEs was assessed and compared among age groups. Severe cases were observed in 20.5% of all cases and were more common in the elderly (30.4%) than in adults (15.8%) and children/adolescents (10.3%, p<0.001). In addition, the rate of hospitalization after ED visits were significantly higher in the elderly (42.8%) than in adults (21.8%) and

**Table 2. Types and severity of ADE-related ED visits.**

| | Total | Age group | | | |
|---|---|---|---|---|---|
| | **(n = 2,104)** | **Children/adolescents** | **Adults** | **Elderlies** | ***P*-value** |
| | | **(n = 242)** | **(n = 1,088)** | **(n = 774)** | |
| **Male** | 829 (39.4%) | 86 (35.5%) | 411 (37.8%) | 322 (41.6%) | 0.128 |
| **Causes** | | | | | |
| Side effect | 1,350 (72.5%) | 252 (62.5%) | 484 (70.7%) | 614 (79.3%) | <0.001 |
| Allergy | 254 (13.6%) | 109 (27.1%) | 105 (15.3%) | 40 (5.2%) | <0.001 |
| Overdose | 136 (7.3%) | 9 (2.2%) | 40 (5.8%) | 87 (11.2%) | <0.001 |
| Non-allergic hypersensitivity | 61 (3.3%) | 24 (6.0%) | 26 (3.8%) | 11 (1.4%) | <0.001 |
| Secondary effect | 37 (2.0%) | 7 (1.7%) | 17 (2.5%) | 13 (1.7%) | 0.505 |
| Drug interaction | 17 (0.9%) | 2 (0.5%) | 3 (0.4%) | 12 (1.6%) | 0.058 |
| Undetermined | 88 (4.7%) | 27 (6.7%) | 41 (6.0%) | 20 (2.6%) | 0.001 |
| **Severity** | | | | | <0.001 |
| Mild | 464 (22.1%) | 40 (16.5%) | 271 (24.9%) | 153 (19.8%) | |
| Moderate | 1,208 (57.4%) | 177 (73.1%) | 645 (59.3%) | 386 (49.9%) | |
| Severe | 432 (20.5%) | 25 (10.3%) | 172 (15.8%) | 235 (30.4%) | |
| **Hospitalization, n (%)** | 613 (29.1%) | 45 (18.6%) | 237 (21.8%) | 331 (42.8%) | <0.001 |
| **Duration of hospitalization, days** | 12.5 ± 27.4 | 7.3 ± 20.1 | 12.8 ± 33.9 | 13.0 ± 22.4 | 0.399 |
| **Death** | 24 (1.1%) | 0 (0%) | 8 (0.7%) | 16 (2.1%) | 0.007 |

ADE, adverse drug event; ED, emergency department

children/adolescents (18.6%, p<0.001). Multivariate logistic regression analysis revealed that old age (OR, 1.792; 95% CI, 1.397–2.297); male gender (OR, 1.275; 95% CI, 1.015–1.603); comorbidities such as chronic liver disease (OR, 4.163; 95% CI, 2.189–7.914), malignancy (OR, 4.032; 95% CI, 2.973–5.498), chronic kidney disease (OR, 4.024; 95% CI, 2.470–6.554), and diabetes (OR, 1.600; 95% CI, 1.211–2.113) were risk factors for severe ADEs (Table 3). In total, 24 ADE-related deaths (1.1%) were noted, mostly in the elderly (n = 16) and adults (n = 8). The mortality rate was higher for the elderly (1.8%) than for adults (0.6%) and children/adolescents (0.0%, p = 0.007).

## Causative drugs of ADEs

Analysis of causative drugs revealed that 271 drugs were implicated in overall ADE-related ED visits (Fig 2). By the 2nd level of subtherapeutic groups according to the ATC classification

**Table 3. Logistic regression analysis of the risk factors for severe ADE.**

| | Mild to moderate | Severe | Univariate | | | Multivariate | | |
|---|---|---|---|---|---|---|---|---|
| | **(n = 1,672)** | **(n = 432)** | **OR** | **95% CI** | ***P*-value** | **OR** | **95% CI** | ***P*-value** |
| **Male** | 623 (37.3%) | 206 (47.7%) | 1.535 | 1.240–1.900 | <0.001 | 1.275 | 1.015–1.603 | 0.037 |
| **Age ≥65 years** | 539 (32.2%) | 235 (54.4%) | 2.508 | 2.022–3.110 | <0.001 | 1.792 | 1.397–2.297 | <0.001 |
| **Comorbidities** | | | | | | | | |
| Diabetes | 329 (19.7%) | 151 (35.0%) | 2.194 | 1.740–2.865 | <0.001 | 1.6 | 1.211–2.113 | 0.001 |
| Hypertension | 375 (22.4%) | 141 (32.6%) | 1.676 | 1.329–2.113 | <0.001 | 1.038 | 0.787–1.369 | 0.794 |
| Chronic kidney disease | 34 (2.0%) | 44 (10.2%) | 5.463 | 3.445–8.663 | <0.001 | 4.024 | 2.470–6.554 | <0.001 |
| Chronic liver disease | 21 (1.3%) | 24 (5.6%) | 4.625 | 2.549–8.390 | <0.001 | 4.163 | 2.189–7.914 | <0.001 |
| Malignancy | 120 (7.2%) | 102 (23.6%) | 3.998 | 2.993–5.340 | <0.001 | 4.032 | 2.973–5.468 | <0.001 |

ADE, adverse drug event; OR, odds ratio; CI, confidence interval

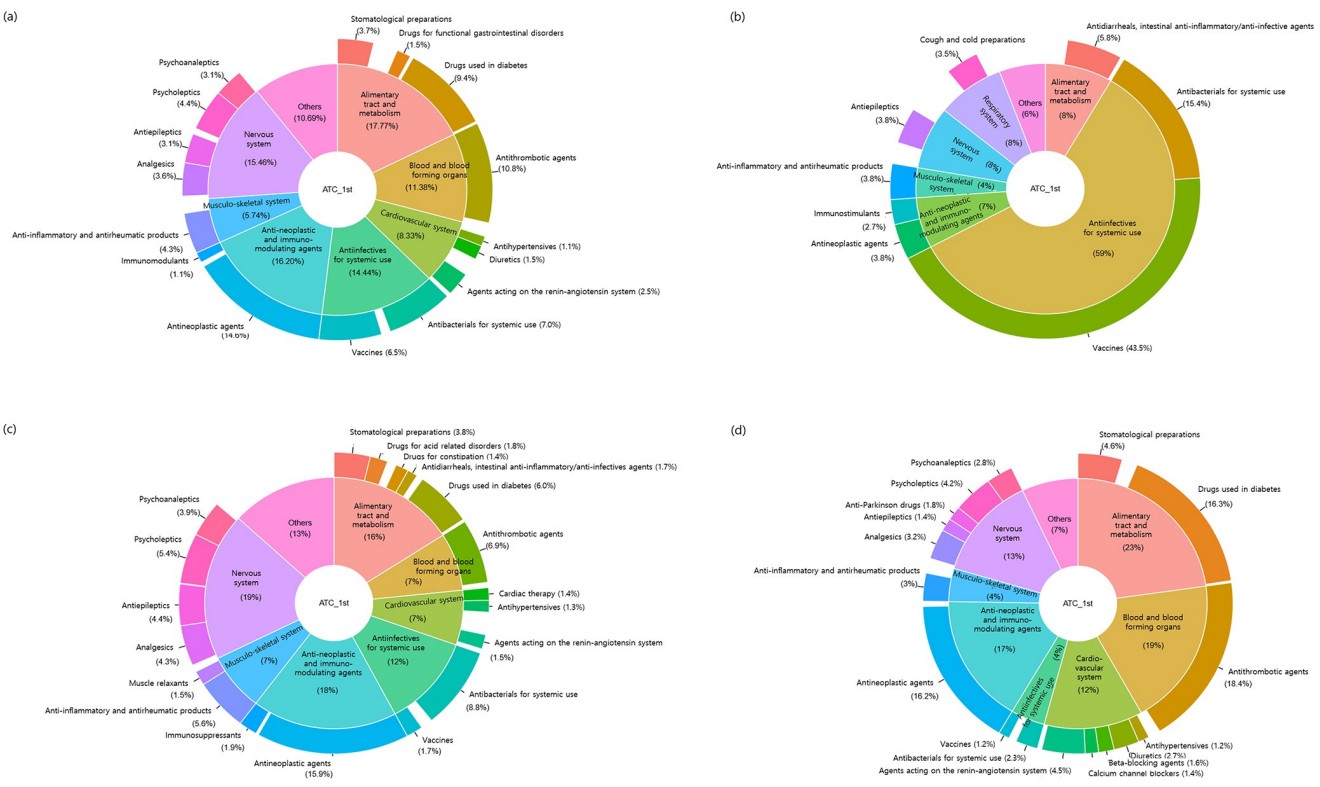

**Fig 2. Common causative drugs of ADE-related ED visits.** (a) Total, (b) children/adolescents, (c) adults, and (d) the elderly.

system (Fig 2a), antineoplastic drugs (L01) were the most common (14.6%) causative drug, followed by antithrombotic agents (B01, 10.8%), drugs used in diabetes (A10, 9.4%), antibacterials for systemic use (J01, 7.0%), vaccines (J07, 6.5%), psycholeptics (N05, 4.4%), anti-inflammatory and anti-rheumatic drugs (M01, 4.3%), stomatological preparations (A01, 3.7%), analgesics (N02, 3.6%), antiepileptics (N03, 3.1%), and psychoanaleptics (N06, 3.1%) in all subjects with ADEs.

Causative drugs varied substantially according to age group. In children/adolescents (Fig 2b), vaccines (J07, 43.5%) were the most common cause of ADEs leading to ED visits, followed by antibacterials for systemic use (J01, 15.4%), antidiarrheal, intestinal anti-inflammatory/ anti-infective agents (A07, 5.8%), and cough and cold preparations (R05, 3.5%). In the adult group (Fig 2c), antineoplastic agents (L01, 15.9%) were the most common causative drugs of ADE-related ED visits, followed by antibacterials for systemic use (J01, 8.8%), antithrombotic agents (B01, 6.9%), drugs used in diabetes (A10, 6.0%), anti-inflammatory and antirheumatic products (M01, 5.6%), and psycholeptics (N05, 5.4%). In contrast, the most frequent causative drugs of ADEs in the elderly group (Fig 2d) were antithrombotic agents (B01, 18.4%), drugs used in diabetes (A10, 16.3%), antineoplastic agents (L01, 16.2%), and agents acting on the renin-angiotensin system (C09, 4.5%). S1 Table presents the causative drugs according to the 5th level of ATC, which is a common generic drug or chemical substance, according to age group.

## Preventability of ADEs

Preventable cases comprised 15.5% (n = 327) of total ADEs (Table 4). Preventable cases were more common in the elderly (18.7%) than in adults (15.1%) or children/adolescents (7.4%,

**Table 4. Preventability of ADE.**

| Criteria of preventability | Total (n = 327) | Age group | | | P-value |
|---|---|---|---|---|---|
| | | Children/ Adolescents (n = 18) | Adults (n = 164) | Elderly (n = 145) | |
| Inappropriate dose, route, or frequency of administration | 158 (44.3%) | 5 (25.0%) | 63 (35.8%) | 90 (56.2%) | <0.001 |
| Poor compliance | 92 (25.8%) | 4 (20.0%) | 63 (35.8%) | 25 (15.6%) | 0.003 |
| Inappropriate for clinical condition | 32 (9.0%) | 3 (15.0%) | 17 (9.6%) | 12 (7.5%) | 0.923 |
| History of allergy or hypersensitivity | 29 (8.2%) | 3 (15.0%) | 22 (12.5%) | 4 (2.5%) | 0.024 |
| Drug interaction | 28 (7.9%) | 2 (10.0%) | 9 (5.1%) | 17 (10.6%) | 0.029 |
| TDM or lab test required but not performed | 9 (2.5%) | 2 (10.0%) | 1 (0.6%) | 6 (3.8%) | 0.050 |
| Toxic drug level | 8 (2.3%) | 1 (5.0%) | 1 (0.6%) | 6 (3.8%) | 0.060 |

ADE, adverse drug event; TDM, therapeutic drug monitoring

p<0.001). Regarding the criteria for determination of preventability, inappropriate dose, route, or frequency of administration was the most common factor (48.3%), followed by poor compliance (28.1%), inappropriate for clinical condition (9.8%), history of allergies or hypersensitivity (8.9%), and drug interactions (8.6%). Moreover, the criteria for preventability varied significantly, depending on the age group. Inappropriate dose, route, or frequency was the most frequent cause for preventability in all age groups. Poor compliance was relatively common in adults, and a history of allergies or hypersensitivity was less common in the elderly population.

Afterward, we analyzed the causative drugs for preventable ADE cases. According to the 2nd level of ATC code (therapeutic subgroup), drugs used in diabetes were the most common drugs (51.2%), followed by antithrombotic agents (10.3%), psycholeptics (8.8%), antiepileptics (3.5%), analgesics (2.4%), thyroid therapy drugs (2.4%), antihypertensives (2.4%), psychoanaleptics (1.8%), agents acting on the renin-angiotensin system (1.8%), and stomatological preparations (1.8%). S2 Table summarizes the common causative drugs in preventable ADEs leading to ED visits across age groups.

## Discussion

In this multicenter retrospective observational study, we evaluated the prevalence and characteristics of ADEs leading to ED visits by analyzing data from EMRs and the NEDIS database in three university hospitals in Korea. We observed that ADE-related ED visits comprised 3.5% of total ED visits. Subjects with ADE-related ED visits were more likely to be female and older. Side effects were the most common cause of ADE among all age groups; however, overdose was more frequent in the elderly, and allergies or hypersensitivity reactions were more common in children/adolescents. Serious ADEs occurred predominantly in the elderly population. Moreover, preventable cases were more common in the elderly due to inappropriate dose, route, or frequent administration. Causative drugs of ADE differed substantially according to age group. We believe that the strength of this study is that it analyzed many cases of ED visits and provided critical insight into the features of ADE by comparing among age groups.

To date, this is the first pharmacoepidemiologic study to evaluate the prevalence and characteristics of ADE-related ED visits in Korea. We retrospectively reviewed the medical records of all participants who were visited the ED during the study periods. ADE-related ED visits comprised of 3.5% of the total ED visits and were estimated to constitute 35.4 (95% CI, 34.2–37.2) per 1,000 ED visits. The prevalence of ADE-related ED visits was varied between studies. Lee et al. from Korea evaluated ADE-related ED visits in a single hospital by screening the

existence of specific diagnostic codes indicating ADEs, which ranged from Y44 to Y45 in EMRs [16]. They reported that ADE-related ED visits only comprised 0.22% of total ED visits, which is a substantially lower percentage than that observed in our study. The low prevalence of ADE-related ED visits in Lee's study might become from screening ADEs with diagnostic codes. Hohl et al. reported that ADE-related ED visits were underreported in EMRs, and less than 10% of ADE-related ED visits documented the ICD-10 codes indicating ADEs in medical records [17]. The prevalence of ADE-related ED visits observed in our study were more comparable to the results of previous studies in other countries with similar study designs such as definition, inclusion, and/or exclusion criteria of ADE [6–9]. Compared to ED visits with non-ADE causes, ADE-related ED visits required hospitalization more frequently. Despite of limitation from retrospective design, we conducted a multi-center and multi-disciplinary study with trained nurses, pharmacists, allergy specialists as well as ED physicians. We also used the comprehensive classification system and causality assessment tool. We retrospectively reviewed the medical records of all patients who visited ED during the study period and, if needed, also reviewed the medical records before and after the ED visits. These well-defined systematic approaches may give us that our findings will reflect the reality of ED visits with ADE better.

The findings from this study confirmed that ADEs are more common and serious in elderly individuals, as reported previously [18–22]. Compared to children and adults, more elderly patients with ED visits due to ADEs required hospitalization. The reasons for which the elderly population is more prone to the development of ADEs, and serious outcomes are unclear yet. Elderly people are having multiple comorbidities and are taking multiple medications simultaneously [23, 24]. The polypharmacy in the elderly was very high at 83.5% in Korea [25] and 81% in Taiwan [26]. In the western, polypharmacy in the elderly was 39.7% in the U.S. [8] and 33% in Finland [27], which was relatively lower than that in the East, but it is an also important medical and socioeconomic issue. Additionally, physiologic changes and altered drug metabolism in old age may expose elderly individuals to an increased risk of ADEs [28]. Our study revealed that ADE-related ED visits were significantly higher in patients with comorbidities such as diabetes, chronic kidney disease, or chronic liver disease, suggesting polypharmacy in the elderly. ADEs such as side effects or overdose more commonly occurred in the elderly. Among them, preventable ADEs such as 'inappropriate dose, route or frequency of administration' or drug interaction were more frequent in the elderly. Therefore, it can be postulated that combined interactions of polypharmacy and frailty physiologic changes might pose the elderly prone to ADE-related ED visits.

A notable finding of this study is that approximately one of seven ADEs (15.5%) were preventable. Antithrombotic agents (e.g., warfarin, clopidogrel, and aspirin), drugs used in diabetes (e.g., insulin and oral hypoglycemic agents), and antineoplastic agents were common drugs in preventable ADE related ED visits. In the elderly, 'inappropriate dose, route or frequency of administration' or drug interaction were frequent compared to that of young or adults. Diabetic drugs and antithrombotic drugs were the most common causative drugs in elderly subjects. These findings were consistent with other studies conducted in the U.S [8], Canada [29, 30], and Italy [31, 32]. The MEREAFaPS study group also reported that antiplatelet agents, anticoagulants, and antidiabetics were common causes of ADE-related ED visits in Italy. It means that drugs for chronic medical conditions are the main causes of ADE-related ED visits and a considerable portion of these are preventable. Therefore, coordinated efforts to develop the strategies for managing common preventable ADEs in the elderly should be needed [33].

The proportion of ADE-related ED visits was also higher in the female gender. Differences in the pattern of drug use might be a factor for the gender difference, but female predominance persisted after adjusting the pattern of drug use [22]. The causes of this finding were unclear

yet, but genetic predisposition [34], differences in metabolizing capacity of cytochrome enzymes might be related to the gender predisposition of ADR [35]. The female dominance of ADE from neurologic or psychiatric drugs and diuretics was also observed in both our and other studies [36–38]. In addition, male predominance of ADE from antineoplastic agents and antithrombotic agents was also observed in both our and other studies [21, 22]. Therefore, this discrepancy of ADE-related ED visits rendered tailored medication prescription and instruction according to gender.

Pharmacovigilance studies based on ED visits are useful to monitor both acute and serious ADEs occurring in the community. Since most drugs are consumed outside hospitals, epidemiologic studies on ADEs in hospitalized patients have limited value for evaluating various adverse reactions in real life. Thus, ADE-related ED visits are a crucial indicator for estimating the health issues and consequent economic burden of ADEs. In the US, a nationwide database has been generated to estimate and monitor adverse drug reactions and prospectively collect data on ADE-related ED visits. Since 2002, the Food and Drug Administration, the Center for Disease Control, and CPSC have cooperated to establish the National Electronic Injury Surveillance System (NEISS)-Cooperative Adverse Drug Event Surveillance (CADES), a system for monitoring harmful drug cases [39]. By using this nationwide registry, ADE-related ED visits are actively monitored [33, 40–42]. We propose that modification of the NEDIS system to incorporate data on ED visits due to ADEs is warranted, as this will enable continuous and comprehensive monitoring of ADEs in a nationwide approach. The employment of a nationwide pharmacovigilance system for ADE-related ED visits will enable a rapid and effective assessment of information on drug safety and adverse effects.

This study had several limitations. First, given that we retrospectively reviewed the medical records of subjects with ED visits, there was limited information regarding clinical presentation, previous history of drug hypersensitivity, and medications used outside the hospital. Some cases might not have been recorded in the EMR despite the apparent ADE-related ED visits. It is difficult to distinguish retrospectively the ADEs from the worsening of the symptoms and signs of the underlying disease. Therefore, causality and causative drugs were occasionally challenging to determine. To overcome this lack of information, we utilized every database source available, including EMRs of the ED, outpatient department, and hospitalization; the NEDIS; and the clinical data warehouse from each hospital. Second, the results of the multicenter retrospective study may be affected by the different systems in each hospital and discrepancies between researchers. To minimize the heterogeneity of the participating institutions, we used the same database of ED visits (NEDIS). Further, the review process and assessment of major outcomes were based on standardized study protocols and consistent definitions. Third, we analyzed ADE cases visiting the ED at only three university hospitals. Thus, the findings of this study may not represent the overall nationwide features of ADEs. It is possible that subjects with more serious ADEs visited or were transferred to the ED at university hospitals from secondary hospitals or clinics. To assess the nationwide prevalence and features of ADEs, it will be necessary to analyze data from the sentinel centers from each region of the nation.

## Conclusion

The increase in drug exposure has led to a rapid increase in the occurrence of adverse drug events (ADEs). Although ADE-related ED visits are a critical issue related to medication safety, the incidence, and epidemiologic characteristics of ADEs leading to ED visits have not been sufficiently studied. To date, this is the first pharmacovigilance study to evaluate ADE-related ED visits, and it is also meaningful that it has been carried out as a multi-center and multi-

disciplinary approach. The prevalence of ADE in ED visits was common in Korea and higher in older adults and females. Many cases of ADEs were preventable and predictable. A further prospective study is needed to evaluate the nationwide burden of ADE leading to ED visits.

## Supporting information

**S1 Table. Causative drugs of ADE related ED visits according to age group.** ADE, adverse drug event; ED, emergency department.
(DOCX)

**S2 Table. Causative drugs of preventable cases of ADE related ED visits (classified by the therapeutic main group).** ADE, adverse drug event; ED, emergency department.
(DOCX)

## Author Contributions

**Conceptualization:** Min-Gyu Kang, Ju-Yeun Lee, Jae-Woo Jung, Hye-Ran Yoon, Hye-Kyung Park, Sang-Heon Kim.

**Data curation:** Min-Gyu Kang, Ju-Yeun Lee, Sung-Il Woo, Kyung-Sook Kim, Jae-Woo Jung, Tae Ho Lim, Ho Joo Yoon, Chan Woong Kim, Sang-Heon Kim.

**Formal analysis:** Min-Gyu Kang, Ju-Yeun Lee, Jae-Woo Jung, Hye-Kyung Park, Sang-Heon Kim.

**Funding acquisition:** Sang-Heon Kim.

**Investigation:** Min-Gyu Kang, Sung-Il Woo, Kyung-Sook Kim, Jae-Woo Jung, Tae Ho Lim, Ho Joo Yoon, Chan Woong Kim, Sang-Heon Kim.

**Methodology:** Min-Gyu Kang, Ju-Yeun Lee, Jae-Woo Jung, Sang-Heon Kim.

**Project administration:** Sang-Heon Kim.

**Supervision:** Sang-Heon Kim.

**Validation:** Min-Gyu Kang.

**Visualization:** Min-Gyu Kang.

**Writing – original draft:** Min-Gyu Kang.

**Writing – review & editing:** Ju-Yeun Lee, Jae-Woo Jung, Ho Joo Yoon, Hye-Ran Yoon, Hye-Kyung Park, Sang-Heon Kim.

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
