## [Decision Letter · Decision Letter 0]

11 Feb 2022

PONE-D-21-39781Adverse drug events leading to emergency department visits: A multi-center observational studyPLOS ONE

Dear Dr. Kim,

Thank you for submitting your manuscript to PLOS ONE. After careful consideration, we feel that it has merit but does not fully meet PLOS ONE’s publication criteria as it currently stands. Therefore, we invite you to submit a revised version of the manuscript that addresses the points raised during the review process.

The manuscript is well assessed by the two reviewers. Major revisions are necessary in the present form. See the reviewers' suggestion carefully and respond them appropriately.

We look forward to receiving your revised manuscript.

Kind regards,

Masaki Mogi

Academic Editor

PLOS ONE

Journal Requirements:

2. In the ethics statement in the manuscript and in the online submission form, please provide additional information about the patient records used in your retrospective study, including: a) whether all data were fully anonymized before you accessed them; b) the date range (month and year) during which patients' medical records were accessed; c) the date range (month and year) during which patients whose medical records were selected for this study sought treatment. If the ethics committee waived the need for informed consent, or patients provided informed written consent to have data from their medical records used in research, please include this information.

“This research was supported by a grant for the Korea Health Technology R&D

Project through the Korea Health Industry Development Institute (KHIDI), funded by the Ministry

of Health & Welfare, Republic of Korea (grant number, HI19C0218); the National Research

Foundation of Korea (NRF) grant funded by the government of Korea (grant number, 2020R1F1A1069087); and a research grant from the Korea Institute of Drug Safety & Risk

Management (2015-0002).”

Reviewers' comments:

Reviewer's Responses to Questions

**Comments to the Author**

1. Is the manuscript technically sound, and do the data support the conclusions?

Reviewer #1: Yes

Reviewer #2: Partly

2. Has the statistical analysis been performed appropriately and rigorously? 

Reviewer #1: Yes

Reviewer #2: Yes

3. Have the authors made all data underlying the findings in their manuscript fully available?

Reviewer #1: Yes

Reviewer #2: Yes

4. Is the manuscript presented in an intelligible fashion and written in standard English?

Reviewer #1: Yes

Reviewer #2: Yes

5. Review Comments to the Author

Reviewer #1: The study is very interesting for the pharmacovigilance area, as it makes a multicentric retrospective study to assess the frequency of adverse events in emergency admission. As mentioned by the author himself, what we have in the literature today with this objective are studies in a single health unit, thus limiting some data and comparisons. The article was well structured and shows interesting results, which can help in better preventive actions to avoid

medication errors, which was the most important data of the article. I have some suggestions that may provide more support: 1) it would be interesting to demonstrate how many medications, on average, by age, the subjects took, and thus relate to the influence of polypharmacy on reactions and medication errors; 2) it would be interesting for medication errors to show a table of which medications are involved in these errors and the type of error, thus being able to direct public policies for the prevention of medication errors; 3) I believe that the major limitation of the study is that it was retrospective, and thus having to believe that everything was in the medical records, I suggest putting as a limitation - that the data presented may be greater, since the data were extracted from medical records, in hospitals different, different clinical care teams, and some reactions may not have been recorded, or even not identified, since some signs and symptoms are confused with diseases.

Reviewer #2: Title

I suggest to indicate the Country where the study was performed (i.e., a Korean multi-center…)

Abstract

I suggest to use visit instead of admission, seriousness instead of severity, adverse drug event (ADE) instead of side effect, serious instead of severe. Please modify these terms also in the manuscript and tables

Statistical analysis performed by the Authors should be added to the Abstract section (where is the multivariate logistic regression reported in the main manuscript?)

More detailed results should be added to the Abstract section (where are the interesting ORs reported in table 3?)

Methods

Which kind of classification system is used by the Author to describe both suspected (causative) medications and observed ADEs? I suggest to detail in this section both ATC and MedDRA classification systems

Which kind of algorithm/scale is used by the Author to estimate ADE preventability? There are several approaches, i.e. the Schumock and Thornton criteria, that could be described and used by the Authors

Discussion

This section should be more extensive and reader-friendly. Authors should perform a point-by-point (demographic, pharmacological and clinical characteristics) comparison between their results and those already published worldwide (i.e., Korea vs Europe, Korea vs United States, Korea vs Canada, Korea vs Australia, etc.)

Some examples of similar analysis performed in Europe that could be useful in the aforementioned comparison are reported below: PMID: 32327995, PMID: 33708120, PMID: 34358104

I also suggest to add here some points of strengths (or to better discuss the few points of strengths already mentioned)

Which are the clinical implications in conducting this kind of pharmacoepidemiological studies? Why pharmacovigilance and pharmacoepidemiology approaches could be considered a valuable options in the evaluation of drug-safety in a real-life setting such as ED?

Figures

The quality of figures is relatively low, please make them more readable

6. PLOS authors have the option to publish the peer review history of their article (what does this mean?). If published, this will include your full peer review and any attached files.

Reviewer #1: **Yes: **Patricia Moriel

Reviewer #2: No

---

## [Author Response · Author response to Decision Letter 0]

29 Mar 2022

Reviewer 1:

The study is very interesting for the pharmacovigilance area, as it makes a multicentric retrospective study to assess the frequency of adverse events in emergency admission. As mentioned by the author himself, what we have in the literature today with this objective are studies in a single health unit, thus limiting some data and comparisons. The article was well structured and shows interesting results, which can help in better preventive actions to avoid medication errors, which was the most important data of the article. 

Response: Thank you for your positive comments and nice summary. We are submitting a revised manuscript that addresses the concerns raised. A detailed, point-by-point response to these concerns is attached.

I have some suggestions that may provide more support: 

1) it would be interesting to demonstrate how many medications, on average, by age, the subjects took, and thus relate to the influence of polypharmacy on reactions and medication errors; 

Response: We absolutely agree with your point of view in that polypharmacy and taking many medications may be associated with increased risk of adverse drug events (ADE). Unfortunately, since we enrolled subjects visiting to emergency department, it was not possible to analyze the medications or prescriptions by the other clinics or hospitals for each patient. Given that polypharmacy is frequently observed especially in older adults, it is speculated that polypharmacy might affect the development of ADE in older adults. We added this point as a limitation of this study (page 18, line299-304). 

2) it would be interesting for medication errors to show a table of which medications are involved in these errors and the type of error, thus being able to direct public policies for the prevention of medication errors; 

Response: Thank you for your comments. We agree with the reviewer’s comments that medication error is an important part to address in this real-world study on pharmacovigilance. However, since we did not analyse the causative drugs related with medication error, it is regrettable to present this data. Instead, we described and presented the medications related with preventable ADE in supplementary Table S2.

3) I believe that the major limitation of the study is that it was retrospective, and thus having to believe that everything was in the medical records, I suggest putting as a limitation - that the data presented may be greater, since the data were extracted from medical records, in hospitals different, different clinical care teams, and some reactions may not have been recorded, or even not identified, since some signs and symptoms are confused with diseases.

Response: Thank you for your helpful comments, which have substantially improved the quality of our manuscript. We also agree that the real body of ADE cases would be bigger than we estimated in this study. According to your suggestion, we described the limitation of this study in the discussion section (page 21, line 353-357).

Review 2

1. Title

I suggest to indicate the Country where the study was performed (i.e., a Korean multi-center…). 

Response: Thank you for your comments. The title was modified including the country as “Adverse drug events leading to emergency department visits: A multicenter observational study in Korea”.

2. Abstract

1) I suggest to use visit instead of admission, seriousness instead of severity, adverse drug event (ADE) instead of side effect, serious instead of severe. Please modify these terms also in the manuscript and tables.

Response: We appreciate the reviewer’s helpful suggestion. We changed certain terms in the abstract and articles based on your suggestions. The word ‘admission’ was changed to the visits except for cases in which an actual ward is admitted. A total of eight ‘severe’ terms were replaced with ‘serious’. The term ‘severe’ was changed to ‘serious’ in statements conveying the severity of ADE in general. However, just as the CTCAE severity grading scale employs the term ‘severity’ rather than ‘seriousness’, the term severity and severity are interchangeable. The term ‘side effect' appears four times in the original manuscript. We used this term to refer to undesirable response due to pharmacologic property of drugs, such as antihistamine-induced sedation, as a part of type A reactions (overdose, secondary effects and drug interactions). Thus, we reserved the term ‘side effects’ for a certain type of ADE. To avoid confusion regarding this issue, we clarified the methods in detail. 

2) Statistical analysis performed by the Authors should be added to the Abstract section (where is the multivariate logistic regression reported in the main manuscript?).

Response: Thank you for your comment. We added the statistical analysis in the abstract.

3) More detailed results should be added to the Abstract section (where are the interesting ORs reported in table 3?).

Response: Thank you for your helpful comments. We detailed the results of the abstract by adding OR and 95% CI in describing the main findings of the multivariate logistic regression analysis presented in Table 3. 

“Multivariate logistic regression analysis revealed that compared to non- adverse drug event-related cases, adverse drug event-related emergency department visitors were more likely to be female (60.6% vs. 53.6%, p<0.001, OR 1.285, 95% CI 1.025-1.603) and older (50.8 ± 24.6 years vs. 37.7 ± 24.4 years, p<0.001, OR 1.892, 95% CI: 1.397-2.297). Comorbidities such as diabetes, chronic kidney disease, chronic liver disease, and malignancies were also significantly associated with adverse drug event-related emergency department visits.”

3. Methods

1) Which kind of classification system is used by the Author to describe both suspected (causative) medications and observed ADEs? I suggest to detail in this section both ATC and MedDRA classification systems.

Response: Thank you for pointing this out, which we did not fully acknowledge in our original manuscript. In this study, the World Health Organization Adverse Reaction Terminology (WHO-ART) was used to describe ADE instead of MedDRA. For the causative drugs, we used the Anatomical Therapeutic Chemical (ATC) Classification System. We modified the description of the methods used in the classification of ADE and causative drugs (page 8, line 124-129).

2) Which kind of algorithm/scale is used by the Author to estimate ADE preventability? There are several approaches, i.e. the Schumock and Thornton criteria, that could be described and used by the Authors.

Response: As you mentioned, we used and characterized the preventability of ADEs by the Schumock and Thornton criteria. the description for the criteria of preventability was added in lines 144-145 of the ‘main outcomes of ADEs’ section.

4. Discussion

This section should be more extensive and reader-friendly. Authors should perform a point-by-point (demographic, pharmacological and clinical characteristics) comparison between their results and those already published worldwide (i.e., Korea vs Europe, Korea vs United States, Korea vs Canada, Korea vs Australia, etc.). Some examples of similar analysis performed in Europe that could be useful in the aforementioned comparison are reported below: PMID: 32327995, PMID: 33708120, PMID: 34358104.

Response: We appreciate the reviewer’s helpful suggestion. We changed the discussion part in the order of incidence, elderly, and female. In addition, our study results were compared with other studies in the order of domestic and foreign studies. According to your comments, we further reviewed the pharmacoepidemiologic studies in Italy and described the summarisez results of the Italian study in the discussion section (page 19, line 320-323).

I also suggest to add here some points of strengths (or to better discuss the few points of strengths already mentioned). Which are the clinical implications in conducting this kind of pharmacoepidemiological studies? Why pharmacovigilance and pharmacoepidemiology approaches could be considered a valuable options in the evaluation of drug-safety in a real-life setting such as ED?

Response: We absolutely agree with the reviewer’s comments that the pharmacovigilance and pharmacoepidemiology approaches offer great opportunity to suggest what is still needed to improve drug safety around the world. According to the reviewer’s suggestion, we added the strength and importance of the pharmacoepidemiologial study of ADE in real world in lines 319-333 and the conclusion.

5. Figures

The quality of figures is relatively low, please make them more readable.

Response: The low resolution of the pictures is due to PLOS one's e-submission system. We made the figures through R software with high resolution. Perhaps, when the paper is published, you will be able to see high-resolution pictures. Following this, we attached the PDF version of the high-resolution figures.

---

## [Decision Letter · Decision Letter 1]

19 Apr 2022

PONE-D-21-39781R1Adverse drug events leading to emergency department visits: A multicenter observational study in KoreaPLOS ONE

Dear Dr. Kim,

Thank you for submitting your manuscript to PLOS ONE. After careful consideration, we feel that it has merit but does not fully meet PLOS ONE’s publication criteria as it currently stands. Therefore, we invite you to submit a revised version of the manuscript that addresses the points raised during the review process.

There are small minor revision in the present form. See the reviewers' comments and respond them.

We look forward to receiving your revised manuscript.

Kind regards,

Masaki Mogi

Academic Editor

PLOS ONE

Journal Requirements:

Reviewers' comments:

Reviewer's Responses to Questions

**Comments to the Author**

1. If the authors have adequately addressed your comments raised in a previous round of review and you feel that this manuscript is now acceptable for publication, you may indicate that here to bypass the “Comments to the Author” section, enter your conflict of interest statement in the “Confidential to Editor” section, and submit your "Accept" recommendation.

Reviewer #1: All comments have been addressed

Reviewer #2: All comments have been addressed

2. Is the manuscript technically sound, and do the data support the conclusions?

Reviewer #1: Yes

Reviewer #2: Yes

3. Has the statistical analysis been performed appropriately and rigorously? 

Reviewer #1: Yes

Reviewer #2: Yes

4. Have the authors made all data underlying the findings in their manuscript fully available?

Reviewer #1: Yes

Reviewer #2: Yes

5. Is the manuscript presented in an intelligible fashion and written in standard English?

Reviewer #1: Yes

Reviewer #2: Yes

6. Review Comments to the Author

Reviewer #1: The authors punctually answered the doubts of the reviewers. Most of the suggestions were accepted, some due to the impossibility of obtaining the data were placed as limitations of the study. The new version of the article is clearer, and despite the limitations, which now appear in the article, the results are important mainly because it is a multicentric study.

Reviewer #2: Please revise the following sentence: "The MEREAFaPS study group also reported that antiplatelet

321 agents, anticoagulants, and antidiabetics were drugs such as antiplatelet agents and

322 anticoagulants were a common cause of ADE-related ED visits in Italy"

7. PLOS authors have the option to publish the peer review history of their article (what does this mean?). If published, this will include your full peer review and any attached files.

Reviewer #1: No

Reviewer #2: No

---

## [Author Response · Author response to Decision Letter 1]

23 Jul 2022

Manuscript PLOS-D-21-39781R1

Response to reviewers

Thank you for giving us the opportunity to submit a revised draft of the manuscript “adverse drug events leading to emergency department visits: A multicenter observational study in Korea” for publication in the PLOS One. We appreciate the time and effort that you and the reviewers dedicated to providing feedback on our manuscript and are grateful for the insightful comments on and valuable improvements to our paper. We have incorporated most of the suggestions made by the reviewers. Those changes are highlighted within the manuscript. Please see below, in red, for a point-by-point response to the reviewers’ comments and concerns. All page numbers refer to the revised manuscript file with tracked changes. 

Reviewers’ Comments to the Authors:

Reviewer 2: Please revise the following sentence: “MEREAFaPS study group also reported that antiplatelet agents, anticoagulants, and antidiabetes were drugs such as antiplatelet agents and anticoagulants were a common cause of ADE-related ED visits in Italy”

Author response: Thank you for pointing this out. The reviewer is correct, and we revised the sentence following as “The MEREAFaPS study also reported that antiplatelet agents, anticoagulants and anti-diabetes were also common causes of ADE-related ED visits in Italy”

---

## [Editor Report · Decision Letter 2]

26 Jul 2022

Adverse drug events leading to emergency department visits: A multicenter observational study in Korea

PONE-D-21-39781R2

Dear Dr. Kim,

We’re pleased to inform you that your manuscript has been judged scientifically suitable for publication and will be formally accepted for publication once it meets all outstanding technical requirements.

Kind regards,

Masaki Mogi

Academic Editor

PLOS ONE
---

## [Editor Report · Acceptance letter]

8 Sep 2022

PONE-D-21-39781R2 

Adverse drug events leading to emergency department visits: A multicenter observational study in Korea 

Dear Dr. Kim:

I'm pleased to inform you that your manuscript has been deemed suitable for publication in PLOS ONE. Congratulations! Your manuscript is now with our production department. 

Kind regards, 

on behalf of

Dr. Masaki Mogi 

Academic Editor

PLOS ONE